# Deep Learning Based Feature Selection and Ensemble Learning for Sintering State Recognition

**DOI:** 10.3390/s23229217

**Published:** 2023-11-16

**Authors:** Xinran Xu, Xiaojun Zhou

**Affiliations:** 1School of Computer Science and Engineering, Central South University, Changsha 410083, China; xinranxu19@gmail.com; 2School of Automation, Central South University, Changsha 410083, China

**Keywords:** sintering state, deep learning, feature selection, binary state transition algorithm, ensemble learning, group decision making

## Abstract

Sintering is a commonly used agglomeration process to prepare iron ore fines for blast furnace. The quality of sinter significantly impacts the blast furnace ironmaking process. In the vast majority of sintering plants, the judgment of sintering quality still relies on the intuitive observation of the cross section at sintering machine tail by operators, which is susceptible to the external environment and the experience of operators. In this paper, we propose a new sintering state recognition method using deep learning based feature selection and ensemble learning. First, features from the infrared thermal images of sinter cross section at the tail of the sinterer are extracted based on ResNeXt. Then, to eliminate the irrelevant, redundant and noisy features, an efficient feature selection method based on binary state transition algorithm (BSTA) is proposed to find the truly useful features. Subsequently, an ensemble learning (EL) method based on group decision making (GDM) is proposed to recognize the sintering states. Novel combination strategies considering the varying performance of the base learners are designed to further improve recognition accuracy. Industrial experiments conducted at a steel plant verify the effectiveness and superiority of the proposed method.

## 1. Introduction

The blast furnace ironmaking process is currently the dominant process worldwide for providing raw materials for steelmaking. It is the main method of modern ironmaking, contributing to more than 95% of the world’s total iron production. In the blast furnace ironmaking industry, iron ore plays an extremely important role as the primary feed material. As high-grade natural iron ore reserves decrease, artificial iron ore is becoming more and more critical [1]. Sintering is the primary way to produce high-quality artificial iron ore from low-grade iron ore, which has been widely used around the world. The production of high-quality sinter is crucial for assuring consistent, stable furnace productivity with a low consumption of reductants.

Estimation of the sintering state plays a vital role in improving the quality of sintering. The state of sintering depends on an important parameter known as burn-through point (BTP). It is a position located on the sinter strand, where the mixed materials thoroughly burn for the first time [2]. By judging the BTP, the sintering states can be recognized. The accurate recognition of sintering state is the guarantee of high-quality and high-yield sinter production, which is of great significance in improving sintering productivity and preventing equipment damage. Nowadays, many sintering plants still rely on experienced operators to observe the sintering machine tail to recognize sintering states with their knowledge and experience. Obviously, this method has many drawbacks. Manual judgment is often affected by many factors such as working experience, working environment, physical stamina, and emotions, resulting in potential errors.

Different from the traditional methods relying on operator judgments, methods based on computer vision depend on the sintering machine tail sectional images to recognize sintering states. Over the years, there has been a remarkable and rapid evolution in vision-based object recognition and is applied in many fields. Significant contributions have been made in various studies such as reconstructing and recognizing human motion with 3D motion analysis [3], supporting disabled individuals through computer-based gesture recognition [4], and exploring novel applications like merging anime with face recognition technology [5]. Similarly, computer vision-based recognition techniques are widely used in the iron ore sintering industry. A series of intelligent modeling methods based on computer vision are established for sintering state recognition. Sun T Q et al. proposed a online sintering quality prediction method based on machine vision and artificial neural network (ANN) [6]. Liu et al. proposed a BTP prediction system based on gradient boosting decision tree (GBDT) algorithm and decision rules [7]. A fuzzy neural network prediction model is presented by Wang et al. [8], in which the strand velocity determines the final model. Li Jiangyun et al. used generative adversarial network (GAN) to expand the sample of sintering data set, and proposed a classification model combining attention mechanism and ResNet [9].

However, many of the proposed methods for sintering state recognition still rely on conventional machine learning methods to extract features from images. In view of the digital image processing technology, a series of features that can comprehensively reflect the sintering state are extracted manually from the images of sinter cross section at the machine tail, such as the average brightness of the red fire layer. This method requires a considerable amount of engineering skills, domain expertise and effort. Deep learning can solve the drawbacks of manual feature extraction by automatically discovering the representations needed for recognition [10,11,12]. However, deep neural networks usually yield a large number of extracted features, including features that are irrelevant, redundant or even noisy for sintering state estimation [13]. These features can bring about high computational complexity and poor learning performance. In addition, it is difficult to determine the best model for recognition tasks without sufficient information, so the recognition accuracy can not be effectively guaranteed with a single model.

In order to deal with the above issues, a novel sintering state recognition method using deep learning based feature selection and ensemble learning is proposed in this paper. A deep neural network ResNeXt pre-trained on ImageNet is used to extract features from infrared thermal images of sinter cross section at the tail of the sintering machine. For the features extracted by deep learning, a feature selection method based on the binary state transition algorithm (BSTA) with a novel training strategy based on feature decomposition is proposed to select the optimal subset of features, which can greatly increase computational efficiency and build better generalization models. Finally, a new ensemble learning (EL) method is proposed to recognize sintering states based on the obtained optimal subset. Considering the limitations of an individual learner and shortcomings of the current EL schemes such as majority voting, the framework of group decision making (GDM) is introduced, which aims to find an optimal alternative considering various suggestions of decision-makers [14]. The main contributions of the proposed method are as follows:•A feature extractor based upon ImageNet-pretrained ResNeXt50 is used to automatically extract fixed features from images of sinter cross section at sintering machine tail.•An efficient feature selection method using the binary state transition algorithm (BSTA) and feature decomposition strategy is proposed to eliminate features that are irrelevant, redundant or even noisy, which not only reduces the difficulty of training but also enhances recognition accuracy.•An ensemble learning framework based on group decision making (GDM) is put forward to further improve recognition accuracy, where new combination strategies are introduced to efficiently fuse base learners.

The remainder of the paper is organized as follows. The industrial background is introduced in Section 2. Section 3 describes the proposed method in detail. In Section 4, experiments are conducted to verify the effectiveness of the proposed method. Section 5 concludes this paper.

## 2. Background

### 2.1. Sintering Process and Problem Description

Iron ore sintering is a crucial step in the ironmaking process [15]. Its primary goal is to produce a material with specific thermal, mechanical, physical, and chemical properties suitable for feeding into the blast furnace. The sintering process, depicted in Figure 1, involves the application of heat to fine iron ore particles, transforming them into coarser grains [16]. Firstly, various raw materials, including iron ore, coke, fluxes, and recycled sintered ore, are blended in a mixer to create a mixture, which is then transported to the mixture bunker. Subsequently, this mixture is evenly spread on a moving trolley to form a sintering bed [1]. The ignition hood is responsible for igniting the surface of the bed and preserving heat, while a blower located beneath the trolley creates a negative pressure zone below the bed through exhaust. As the trolley slowly moves towards the tail of the sintering machine, combustion progresses downward. Gradually, the raw ore powder transforms into sintered ore with specific particle sizes, becoming a valuable feedstock for subsequent stages of blast furnace ironmaking production.

The recognition of sintering state is a pivotal part in assessing sintering quality. Excessive sintering temperatures and prolonged sintering durations can lead to a decline in the ultimate product performance, a phenomenon known as oversintering. Conversely, when sintering temperatures are too low and sintering time is too short, the product fails to attain the necessary performance criteria, a condition referred to as undersintering. Obviously, different sintering states will result in products of varying quality, and only the normal sintering state is desired. Hence, it is necessary to develop an efficient method for accurately assessing the sintering states to ensure the production of high-quality sintered ore.

### 2.2. Image Acquisition

The tail of the sintering machine marks the conclusion of the sintering process, and it is an important location where a plethora of sintering parameters can be captured from cross-sectional images of the sinter. We deployed the Fluke TiX1000 infrared thermal imager positioned at the viewing port at the end of the sintering machine. This imager boasts a temperature measurement range spanning from −40 °C to 2000 °C and a high-resolution capability of 1024 × 768 pixels. This device was connected to a computer through a fiber optic network for seamless real-time transmission of the collected infrared thermal images. The obtained images of sinter cross section at the tail of the sintering machine are shown in Figure 2.

## 3. Proposed Method

### 3.1. Overview of the Proposed Method

This section illustrates the proposed sintering state recognition method in details. The overall framework of the proposed method is shown on Figure 3. This framework includes three modules: feature extraction, feature selection, and sintering state recognition based on ensemble learning.

Traditional feature extraction methods based on computer vision suffer from drawbacks of high cost and low efficiency. In this paper, the features from the images of sinter cross section at the sintering machine tail are extracted by a feature extractor built on a ResNeXt50 architecture that has been pretrained on the ImageNet dataset using transfer learning. Currently, ImageNet transfer is the most effective and economical choice. Representations from deep networks are generic and can support transfer learning across domains [17,18].

The number of features obtained is often very large after deep learning-based image feature extraction. Such high-dimensional data is likely to contain redundant, irrelevant, or even noisy features, which significantly increase the training difficulty of the recognition model and lead to reduced recognition accuracy. Feature selection can effectively tackle this problem by identifying the optimal features from the original feature set. This paper proposes a new feature selection method using binary state transition algorithm. A novel training strategy based on feature decomposition is introduced to reduce the computation time of BSTA and the number of selected features.

Following the data preprocessing steps of feature extraction and feature selection, the optimal features that encapsulate the key information from the sinter cross section images are obtained. The sintering state is recognized utilizing this feature set. Considering the limitations of a single individual model, this paper proposes a novel ensemble learning method based on group decision making to judge sintering states. Novel base learner combination strategies that integrate multiple evaluation metrics are put forward to address the shortcomings existing in previous ensemble learning methods.

### 3.2. Feature Extraction Based on Deep Learning

Conventional machine learning methods have limitations when it comes to processing raw natural data. In the case of sintering image recognition, traditional techniques of extracting features are able to acquire shallow features including the red fire layer area, the average brightness of the red fire layer, etc. But this method requires careful engineering and considerable domain expertise to manually design a feature extractor that converts the raw data into a suitable representation [10]. In fact, this can be circumvented by deep learning which automatically learns useful features through a generic learning procedure. The ResNet-based architecture is a widely adopted deep neural network in image recognition [19]. By introducing the residual connections to the CNN [20], ResNet is capable of solving the degradation problem in very deep networks and improving model performance. ResNeXt [21] was proposed as a variant of ResNet with the redesigned building blocks. It develops an aggregated transformation strategy combining the block stack strategy of ResNet with group convolution techniques of the Inception architecture. A ResNeXt block follows the split-transform-merge strategy. Instead of performing a single transformation as in the ResNet block, it performs a series of transformations. Highly competitive recognition performance is achieved in this way without increasing the complexity of the model.

In this work, the features from the images of sinter cross section at the sintering machine tail are extracted by a feature extractor built upon a ResNeXt50(32x4d) architecture that has been pretrained on the ImageNet dataset. In experiments conducted on Imagenet datasets, ResNeXt achieved high accuracy with a relatively low number of flops compared to existing models. Using an pretrained network as a fixed feature extractor is a form of transfer learning [22], which leverages the model’s prior learning of general visual features in a large dataset. Such a property can be very important in fields such as iron ore sintering, where the availability of labels involves extended effort and cost for acquisition. By adopting this strategy, the time-intensive training process is circumvented, saving significant time and computational resources while enhancing performance with a strong capability to capture meaningful features. In each layer of the ResNeXt, there is a new representation of the input image by progressively extracting meaningful information. In this work, the final linear layer has been removed and the features from the last average pooling layer are selected. The architecture of the ResNeXt-based feature extractor is shown in Figure 4. The features extracted by the neural network can be called deep features which contain much more information than the shallow features.

### 3.3. Feature Selection Based on BSTA

Feature selection is an important step in selecting the most relevant features from the original set, which helps to enhance a model’s focus on key information and reduce computational costs. For classification tasks, feature section can greatly improve classification accuracy. In this section, a novel feature selection method based on binary state transition algorithm with feature decomposition strategy is proposed.

#### 3.3.1. Overview of Feature Selection

Suppose S is a dataset containing *M* samples and *N* features. Feature selection aims to find the best feature subset that contains n(n⩽N) features that can maximize the classification accuracy while using as few features as possible [23]. In this work, we represent a solution to the feature selection problem using a binary encoding vector denoted as x. The encoding vector is described as follows:(1)$$\begin{array}{c} x=\left(x_{1},x_{2},\ldots ,x_{n}\right),x_{i}\in \left\{0,1\right\},i=1,2,\ldots ,n \end{array}$$
where $x_{i}=1$ means that the *i*th feature is selected, while $x_{i}=0$ indicates that the feature is not selected. For instance, Figure 5 illustrates a 9-dimensional vector x. The solution x=[1,0,0,1,0,1,1,0,0] corresponds to the selection of 1st, 4th, 6th, and 7th features.

The feature selection problem can be represented as the follows:(2)$$\begin{array}{cc} \; & \max\;f_{1}=Acc\left(x\right)\\ \; & \min\;f_{2}={∥x∥}_{0}\\ s.t.\; & x=\left(x_{1},x_{2},\ldots ,x_{n}\right),x_{i}\in \left\{0,1\right\},i=1,2,\ldots ,n,1⩽{∥x∥}_{0}\leq n \end{array}$$
where Acc(x) denotes the classification accuracy of the model established based on the corresponding x. ${∥x∥}_{0}$ represents the number features selected from x.

#### 3.3.2. Feature Selection Based on BSTA and Feature Decomposition

In this study, a feature selection method using binary state transition algorithm (BSTA) is proposed for higher classification accuracy and shorter training time. The state transition algorithm (STA) was firstly proposed by Zhou (the co-author of this paper) [24] for continuous optimization problems, which has demonstrated outstanding performance in real-world applications. STA was inspired by state space representation from control theory. It is an individual-based optimization algorithm whose main idea is to produce candidate solutions using four intelligent search operators. These candidates are evaluated, and the algorithm chooses the current best solution, which is extremely efficient in practical applications for the search for the global optimal solution. In STA, a candidate solution is defined as a state, and changes in solutions are described as state transition. To solve integer optimization challenges, the discrete state transition algorithm (DSTA) is proposed as a discrete variant of STA  [25]. Binary State Transition Algorithm (BSTA) is a novel intelligent optimization method designed for boolean integer optimization problems, representing the binary adaptation of the DSTA.

Four specialized state transformation operators have been introduced in BSTA to generate candidates for both local and global search.

(1)Swap transformation:
(3)$$x_{k+1}=A_{k}^{swap}\left(m_{a}\right)x_{k}$$
where, $A_{k}^{swap}\in R^{n\times n}$ represents a swap transformation matrix, and $m_{a}$ is a swap transformation factor which is a constant integer used to control the maximum number of swaps. Swap transformation is illustrated in Figure 6.(2)Shift transformation:
(4)$$x_{k+1}=A_{k}^{shift}\left(m_{b}\right)x_{k}$$
where, $A_{k}^{shift}\in R^{n\times n}$ is a shift transformation matrix, and $m_{b}$ is a shift transformation factor which is a constant integer used to control the maximum length of the moved position. Shift transformation is illustrated in Figure 7.(3)Symmetry transformation:
(5)$$x_{k+1}=A_{k}^{sym}\left(m_{c}\right)x_{k}$$
where, $A_{k}^{sym}\in R^{n\times n}$ is a symmetry transformation matrix, and $m_{c}$ is a symmetry transformation factor which is a constant integer used to control the maximum length of symmetric sequence. Symmetry transformation is illustrated in Figure 8.(4)Substitute transformation:
(6)$$x_{k+1}=A_{k}^{sub}\left(m_{d}\right)x_{k}$$
where, $A_{k}^{sub}\in R^{n\times n}$ is a substitute transformation matrix, and $m_{d}$ is a substitute transformation factor which is a constant integer used to control the maximum number of substitute. Substitute transformation is illustrated in Figure 9.

Algorithm 1 provides the procedure of binary state transition algorithm. swap(.), shift(.), symmetry(.) and substitute(.) are transformation operator functions. In each iteration, these operator functions are performed to generate the candidate solutions. Best represents the best candidate solution so far. The algorithm continues until the predefined maximum number of iterations is reached, and the termination condition is satisfied. The current best solution is then returned.
**Algorithm 1** Binary State Transition Algorithm.1:**function** BSTA(*s*)2:      **repeat**3:            Best←swap(Best,∗)4:            Best←shift(Best,∗)5:            Best←symmetry(Best,∗)6:            Best←substitute(Best,∗)7:      **until** the termination condition is met8:      Return Best9:**end function**

A hybrid feature selection method called ReliefF-BSTA was proposed in pursuit of higher classification accuracy and lower computational resources [26]. ReliefF-BSTA is a hybrid feature selection method that combines the strengths of the feature weighting algorithm ReliefF and the intelligent optimization approach BSTA. In this method, ReliefF narrows down the search space and provides valuable insights into features, and BSTA searches for the optimal feature subset on the basis of feature ranking and feature weights. ReliefF is a well-known filter-based feature selection method, which seeks the best feature subset by computing the features’ weights [27]. It assigns different weights to features according to the correlations between features and categories. Features with weights greater than a predefined threshold will be selected. In multi-label problems, assuming that the labels of the training dataset are $L=\left\{l_{1},l_{2},\ldots ,l_{n}\right\}$, the ReliefF randomly selects a sample $R_{i}$ from the training dataset. Then it searches for *k* nearest neighbors (called near Hits) of $R_{i}$ with the same label, referred to as $H_{j}\left(j=1,2,\ldots ,k\right)$. It also searches for *k* nearest neighbors (called near Misses) of $R_{i}$ with different label, which is denoted by $M_{j}\left(l\right)\left(j=1,2,\ldots ,k\right)$. This process is iterated a total of *m* times by ReliefF. The weight assigned to feature *X* is updated as follows:(7)$$\begin{array}{cc} W(X)= & W\left(X\right)-\sum_{j=1}^{k}\frac{\mathrm{diff}\left(X,R_{i},H_{j}\right)}{\left(m)(k\right)}\\ & +\sum_{l\notin \;\mathrm{class}\;(R)}\frac{\left[\frac{p(l)}{1-p(class(R))}\sum_{j=1}^{k}\mathrm{diff}\left(X,R_{i},M_{j}\left(l\right)\right)\right]}{\left(m)(k\right)} \end{array}$$
where *m* is the number of iterations and $diff(X,R_{i},R_{j})$ denotes the difference of samples $R_{i}$ and samples $R_{j}$ on feature *X*, which is defined as:(8)$$\begin{array}{c} \mathrm{diff}\left(X,R_{i},R_{j}\right)=\left\{\begin{array}{cc} \frac{|R_{i}\left[X\right]-R_{j}\left[X\right]|}{\max(X)-\min(X)} & X\;\mathrm{is}\;\mathrm{continuous}\;\\ 0 & X\;\mathrm{is}\;\mathrm{discrete}\\ & \mathrm{and}\;R_{i}\left[X\right]=R_{j}\left[X\right]\\ 1 & X\;\mathrm{is}\;\mathrm{discrete}\\ & \mathrm{and}\;R_{i}\left[X\right]\neq R_{j}\left[X\right] \end{array}\right. \end{array}$$

Considering the industrial demand for high efficiency, feature selection, as a crucial step in data preprocessing, should prioritize shorter processing time. Recognizing that it takes a lot of time to traverse the complete dataset with BSTA in each iteration, especially when dealing with large datasets in the feature selection problem, this paper proposes a new training strategy called feature decomposition (FD). The dataset is divided into several parts $s_{1},s_{2},\ldots ,s_{n}$ based on features, and each part represents a sub-dataset containing a portion of the features in the original training set. Training is then performed separately for each sub-dataset and the corresponding solutions $X_{1},X_{2},\ldots ,X_{N}$ are obtained. Upon completing the training of all sub-datasets, all solutions are merged and formed into a new dataset termed the final sub-dataset. To derive the ultimate optimal solution, training is conducted on the final sub-dataset. Figure 10 shows the process of the FD strategy.

Our feature selection method comprises a two-stage process. First, the filter method ReliefF is used to narrow the search space and calculate feature weights and feature rankings. Next, the wrapper method BSTA searches for the best feature subset containing the most useful, relevant, and non-redundant features leveraging the important information provided by ReliefF. Feature decomposition strategy is proposed to greatly reduce both the number of selected features and the overall running time while maintaining high classification accuracy. The pseudocode of our proposed feature selection method is illustrated in Algorithm 2.
**Algorithm 2** Pseudocode of ReliefF-BSTA with feature decomposition.1:**Input:** Dataset *D*2:$\left\{D_{1},D_{2},\ldots ,D_{n}\right\}\leftarrow Decompose\left(D\right)$3:**for** each i∈[1,n] **do**4:      Calculate features ranking and feature weights using ReliefF5:      Get the initial $s_{i}$ solution for $D_{i}$6:      $Best_{i}\leftarrow BSTA\left(s_{i}\right)$7:**end for**8:$s\leftarrow Concat(Best_{1},\ldots ,Best_{n})$9:$Best^{*}\leftarrow BSTA\left(s\right)$10:Return $Best^{*}$

### 3.4. Proposed Ensemble Learning Method Based on Group Decision Making

This section introduces how to perform recognition tasks based on the feature subset obtained by feature selection. Considering the limitations of a single model and the shortcomings existing in traditional ensemble learning methods, we propose a novel ensemble learning method based on group decision making framework with novel base learner combination strategies.

#### 3.4.1. Group Decision Making

It is common for us to consider multiple suggestions when making a decision. The process of considering and integrating recommendations from peers or experts into the decision-making process is referred to as group decision making (GDM). GDM aims to determine the best alternative among a set of options, taking into account recommendations from multiple advisors [28].

In a GDM problem with a total of *m* alternatives, we denote these alternatives as $Q_{1},Q_{2},\ldots ,Q_{m}$, which represent the available options. There are *n* decision criteria, called $C_{1},C_{2},\ldots ,C_{n}$ [29]. A set of *K* decision makers(DMs) $E_{1},E_{2},\ldots ,E_{K}$ are assembled, and each decision maker evaluates every alternative independently according to different criteria. The decision making scores generated from DM *k* constitute the decision matrix $D_{k}$:(9)$$\left[\begin{array}{ccccc} d_{11}^{k} & \ldots & d_{1j}^{k} & \ldots & d_{1m}^{k}\\ \vdots & \ddots & \vdots & \ddots & \vdots \\ d_{i1}^{k} & \ldots & d_{ij}^{k} & \ldots & d_{im}^{k}\\ \vdots & \ddots & \vdots & \ddots & \vdots \\ d_{n1}^{k} & \ldots & d_{nj}^{k} & \ldots & d_{nm}^{k} \end{array}\right]$$
where $d_{i}^{k}\left(i=1,2,\ldots ,m;j=1,2,\ldots ,K\right)$ is called performance rating, which represents the assessment provided by decision maker $E_{k}$ for alternative $Q_{i}$ regarding criterion $C_{j}$. The performance rating $d_{ij}^{k}$ serves as a metric to quantify the extent to which $Q_{i}$ satisfies $C_{j}$ from the perspective of $E_{k}$.

The important weight of each criterion and each decision maker can be denoted as $w_{1},w_{2},\ldots ,w_{n}$ and $W_{1},W_{2},\ldots ,W_{K}$ respectively. Performance ratings and important weights range from 0 to 1. Decision matrices $D_{1},D_{2},\ldots ,D_{k}$ are aggregated with important weights in the decision-making process. In this study, GDM is utilized to combine the results generated by the base learners in ensemble learning.

#### 3.4.2. Ensemble Learning Based on GDM

Ensemble learning is often taken as the embodiment of crowd intelligence in machine learning and typically exhibits superior generalization capabilities when compared to individual models [30,31]. It refers to the process of generating and combining multiple base learners in order to complete a specific task. In this work, a new ensemble learning method based on GDM (GDM-EL) with novel combination strategies is proposed. An ensemble learning method can be seen as a decision-making process, where multiple individual learners act as decision makers to collectively determine the final result.

Under the framework of GDM-EL, base learners can be considered as decision makers (DMs), and various categories can be treated as alternatives [32]. For a multi-classification problem with *m* categories and *k* individual learners, a group of *K* decision makers and a set of *m* alternatives are created. Then the performance ratings generated by each of the *K* decision makers are collected and organized into matrices $D_{k}$. In this paper, we use the membership degree (MD) as the criterion to reflect the performance of the alternatives [33], which refers to the degree to which an element belongs to a particular set or category in fuzzy logic and fuzzy set theory. The decision matrix $D_{K}$ produced by decision maker *k* is described as:(10)$$D_{k}=\left[d_{1}^{k},d_{2}^{k},\ldots ,d_{m}^{k}\right]\;\left(k=1,2,\ldots ,K\right)$$
where the performance rating $d_{i}^{k}\left(i=1,2,\ldots ,m;j=1,2,\ldots ,K\right)$ is a numerical value between 0 and 1 that indicates the extent to which a sample belongs to $Q_{i}$ for decision maker $E_{k}$.

When *K* base learners are used in GDM-EL and they evaluate each category based on the membership degree, the important weight assigned to each of them can be represented as $W=[W_{1},W_{2},\ldots ,W_{K}]$. The weights of decision makers are important to combine multiple decision matrices into a single decision matrix. Among all the combination strategies in ensemble learning, the majority vote is by far the simplest for implementation, which gives every base learner the same weight and derives single ground truth labels from multiple base learners [34]. This voting method treats each voter as a completely equal individual, ignoring the different performance of each base learner. However, the classification ability of each classifier for each category differs from one another. Hence, it is essential to incorporate the classification performance of each base learner as its weight in the final decision, necessitating a quantitative measure of their classification power. Precision, recall, and accuracy are metrics that effectively evaluate the performance of a classification method. In this study, instead of relying on a single metric, the performance is assessed by considering a combination of precision, recall, and accuracy measures. By utilizing these multiple metrics, a more comprehensive evaluation of the base learners’ performance is achieved. The precision, recall and accuracy are presented below.
(11)$$\begin{array}{c} P=\frac{TP}{TP+FP} \end{array}$$
(12)$$\begin{array}{c} R=\frac{TP}{TP+FN} \end{array}$$
(13)$$\begin{array}{c} A=\frac{Cr}{Total} \end{array}$$
where P,R,A are the precision, recall and accuracy; Cr is the number of correct predictions, and Total is the total number of predictions. A class could be identified as a positive class and the rest of the classes belong to negative classes. TP,FP,FN can be found in in Table 1.

Accuracy is the most commonly used index to measure the performance of a classification model on a dataset. The accuracy obtained by classification method *k* is represented as $A_{k}$. Precision and recall are designed for a certain category, which are used to evaluate the model’s performance for that category specifically. For each classification method, if the dataset has a total of *m* categories, we calculate the precision and recall of the classification method *k* for each label, then two vectors can be obtained as ${\left[p_{1},p_{2},\ldots ,p_{m}\right]}^{T},{\left[r_{1},r_{2},\ldots ,r_{m}\right]}^{T}$. In this study, we compute the average of each vector and denote them as $P_{k},R_{k}$, calculated as:(14)$$\begin{array}{c} P_{k}=\frac{1}{m}\sum_{i=1}^{m}p_{i} \end{array}$$(15)$$\begin{array}{c} R_{k}=\frac{1}{m}\sum_{i=1}^{m}r_{i} \end{array}$$
where $p_{i},r_{i}$ represents the precision and recall of classification method *k* for category *i*. For a classification problem with *K* base learners, three vectors of length *K* can be obtained as above, respectively described as $P=\left[P_{1},P_{2},\ldots ,P_{K}\right],R=\left[R_{1},R_{2},\ldots ,R_{K}\right],A=\left[A_{1},A_{2},\ldots ,A_{K}\right]$.

Evaluating the overall performance of a model requires consideration of all relevant metrics. Specifically, precision, recall and accuracy should be appropriately integrated to collectively assign weights to the base learners and achieve the best possible outcome. In this paper, three base learner combination strategies are proposed, which combine P,R,A in three different ways to generate weights for decision makers. The combination strategies are inspired by arithmetic mean, geometric mean, and harmonic mean respectively. The 3 weights for DM *k* are $W_{k}^{1},W_{k}^{2},W_{k}^{3}$, defined as follows:(16)$$W_{k}^{1}=P_{k}+R_{k}+A_{k}$$
(17)$$W_{k}^{2}=P_{k}R_{k}A_{k}$$
(18)$$W_{k}^{3}=\frac{3}{\frac{1}{P_{k}}+\frac{1}{R_{k}}+\frac{1}{A_{k}}}$$

From Equations (Equation 16)–(Equation 18), it is obvious that the larger the value of the three indicators, the greater the weight of this base learner. Finally, the score for each option is based on the ratings from all base learners, and these ratings are combined to derive the final score, considering their respective weights. The alternative $Q_{j}$ with the highest score is the final alternative $Q^{*}$, shown as follows:(19)$$Q^{*}=Q_{\underset{j}{\arg\max}\sum_{k=1}^{K}W_{k}d_{j}^{k}}$$

Our method GDM-EL is also compared with the majority vote EL method in Table 2. It is evident that the majority vote adds the votes of each base learner linearly, while GDM-EL performs a weighted combination of the base learners’ results.

The flowchart for GDM-EL is shown in Figure 11. The dataset is divided into training set, validation set and testing set. Several models are generated on the training set to create base learners. Then the base learners are tested on the validation set to obtain performance indexes based on P,R,A. According to these knowledge, the important weights of the base learners can be acquired. Next, for the testing set, GDM is introduced to fuse outputs from these base learners on the testing set. Base learners give each alternative evaluating scores known as performance ratings, and the decision matrices are structured according to these performance ratings. Finally, the best alternative is chosen based on the weight of base learners and decision matrices.

## 4. Experiments and Results

In order to verify the effectiveness of the recognition method proposed above, we conducted industrial experiments on the infrared thermal images of sinter cross section collected at the tail of the sintering machine. The original data came from a sintering plant in Hunan, China. The experimental results demonstrate the effectiveness and feasibility of the proposed method.

The ImageNet-pretrained ResNeXt50 model is used as a fixed feature extractor to extract features from images captured at the sintering machine tail with PCA to further reduce the dimensionality of the data. The architecture of the ResNext-based feature extractor is shown in Figure 4, where fixed feature representations are extracted. Next, a based feature selection method with feature decomposition strategy chooses the best features from those initially extracted by ResNeXt50. Table 3 shows the detailed parameter settings of the feature selection algorithm based on BSTA. Population represents the total number of iterations. Since the BSTA belongs to the individual-based algorithm, SE represents the number of generated candidate solutions per iteration. (pct, *p*, and *q*) are user-specified parameters to control the generation of the initial solution [26]. The number of sub-datasets during feature decomposition and the total number of rounds for feature section are also presented. Table 4 shows the specific information of the dataset we obtain through feature extraction and feature selection.

In this work, 5 common classification methods are employed as base learners of the EL method to judge the sintering quality based on the dataset described in Table 4. The base learners are Support Vector Machine(SVM) [35], Adaboost [36], Logistic Regression(LR) [37], k-Nearest Neighbor(KNN) [38], Random Forest(RF) [39]. On the basis of the optimal dataset after feature selection, these classical machine learning methods, serving as base learners for ensemble learning, can efficiently and rapidly yield satisfactory recognition results. In this paper, they are employed to ensure the diversity of base learners. Moreover, seven real-world datasets are used to compare the three base learner combination strategies in Equations (Equation 16)–(Equation 18). These public datasets are all available at the UCI Machine Learning Repository, namely: Letter Image Recognition Data, Blocks Classification, Dry Beans Dataset, Musk, ISOLET (Isolated Letter Speech Recognition), Pen-Based Recognition of Handwritten Digits, Waveform Database Generator. Details about these datasets can be found in Table 5. Each dataset is split into three subsets: a training set, a validation set, and a testing set. The training set is used to generate base learners. The validation set is applied to generate the priori knowledge of base learners from which the weights of each base learner can be calculated by three combination strategies. Next, for each sample in the testing set, each decision maker assigns a score for every alternative, and 5 decision matrices can be obtained as Equation (Equation 10). At last, GDM is used to fuse the information and determine the final label, and results are calculated as Equation (Equation 19).

Based on the idea of decision-making, each dataset chooses the best combination strategy according to the classification result on the testing set. Finally, the best combination strategy to generate weights can be selected by voting. The results are listed in Table 6, which presents the classification accuracy (%) for each combination strategy on each dataset. It can be seen from the results that $W^{2}=P_{k}R_{k}A_{k}$ achieves the highest accuracy on more datasets than the other two strategies.

Based on the above results, in our experiment of sintering state recognition, we choose $W^{2}=P_{k}R_{k}A_{k}$ as our combination strategy to assign weights to the base learners. The dataset obtained by feature extraction and feature selection from sintering images (described in Table 4) is divided into training set, validation set and testing set. The weight of each decision maker is calculated based on the validation set with combination strategy $W=P_{k}R_{k}A_{k}$. The weights of base learners for ensemble learning are shown in Table 7.

The final results are calculated as Equation (Equation 19). Figure 12 presents the recognition performance of the base learners, the EL method based on majority vote, and our proposed EL method based on GDM (GDM-EL). The effectiveness of the feature selection method based on BSTA is also evaluated in Figure 12. It is evident in Figure 12 that our proposed method demonstrates outstanding performance in sintering state recognition with a recognition accuracy of 97.579%. Specifically, GDM-EL exhibits superior recognition accuracy compared to both the majority vote and individual base learners. Additionally, our feature selection method based on BSTA proves to be effective in enhancing the recognition performance of both base learns, Majority vote, and GDM-EL. The accuracy of each method increases significantly after feature selection. Figure 13a plots the confusion matrix of GDM-EL method without feature selection, and Figure 13b plots the confusion matrix of GDM-EL method with feature selection. It can be found that for each category, the number of correctly recognized samples is increased after BSTA-based feature selection. Figure 13b shows that the proposed method demonstrates excellent recognition performance for each category.

To further verify the superiority of the proposed method, we also conducted experiments with popular recognition methods based on deep learning. We examined the recognition performance in two settings: (1) training a VGG16 [40], Inception-v3 [41], ResNet50 and ResNeXt50 model from scratch with randomly initialized weights, (2) fine-tuning the ImageNet pre-trained ResNeXt50 model. These settings are commonly used in deep learning and transfer learning for training end-to-end models. In the first setting, we examined VGG16, Inception-v3, ResNet50 and ResNeXt50 trained from random initialization using the sinter cross section images and labels for 200 epochs, with a cosine decay learning rate schedule at a batch size of 64. In the second setting, we initialized ResNeXt50 from the ImageNet weights and fine-tuned using a similar training setup. The recognition results are presented in Table 8, where the accuracy and F1-score are reported. As can be observed, our proposed method achieves the highest accuracy and F1-score. The outcome demonstrates the superior effectiveness of our method compared to popular deep learning models that rely solely on supervised training for sintering state recognition.

Based on the experimental results presented above, it is evident that the proposed method in this paper achieves accurate recognition of the sintering states. Utilizing a feature extractor built upon the ImageNet-pretrained ResNeXt50, relevant features from infrared thermal images are efficiently extracted. Significantly, the BSTA-based feature selection technique and the ensemble learning method based on GDM contribute to a remarkable advancement in enhancing the accuracy of sintering image recognition.

## 5. Conclusions

In this study, a new sintering state recognition method using deep learning-based feature selection and ensemble learning is proposed to improve the quality of sintered products and reduce energy consumption. A pretrained deep neural network ResNeXt50 is used to extract features from the infrared thermal images of the sinter cross section at sintering machine tail. To handle the high-dimensional features, a feature selection method using binary state transition algorithm and feature decomposition strategy is proposed. With the acquired feature subset, a novel ensemble learning method based on group decision making with novel base learner combination strategies is proposed to estimate sintering states.

The results of industrial experiments showed that the proposed method efficiently recognizes the sintering states with an impressive accuracy of 97.579%. Pretrained ResNeXt is an effective architecture for constructing feature extractors. Feature selection method based on BSTA has proved effective in providing a useful feature subset for higher accuracy and efficiency. The ensemble learning method based on GDM achieves higher accuracy than individual learners and conventional EL methods. Additionally, the proposed method for sintering state recognition demonstrates better performance compared with popular deep learning models. In future work, we will explore more evaluating criteria in the decision-making process and use more advanced machine learning algorithms to enhance the interpretability and applicability of our methods. We believe that the impact of our research extends beyond immediate application in industrial image recognition, with the potential to have broader applications in other vision-based recognition tasks and diverse areas like object detection and autonomous systems.

## Figures and Tables

**Figure 1 sensors-23-09217-f001:**
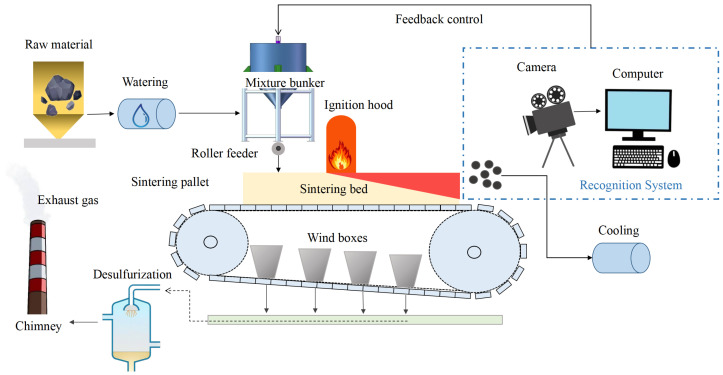
The flowchart of sintering process and feedback control system.

**Figure 2 sensors-23-09217-f002:**
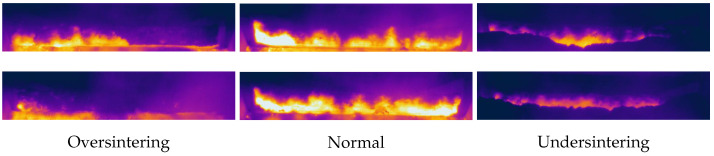
Infrared thermal images of sinter cross section at the tail of the sintering machine.

**Figure 3 sensors-23-09217-f003:**
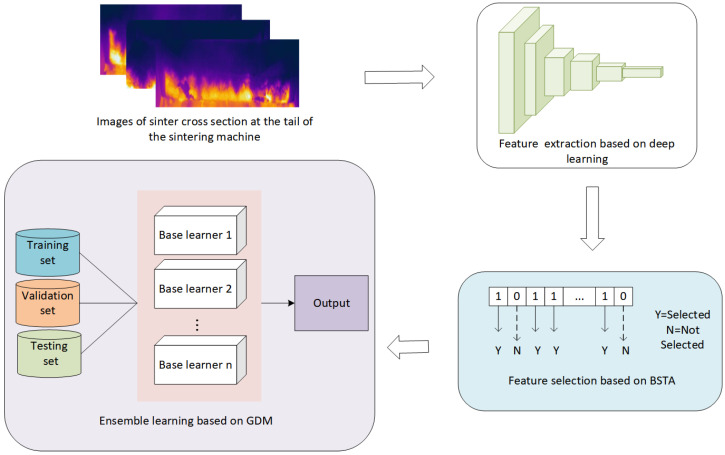
Framework of the proposed method.

**Figure 4 sensors-23-09217-f004:**
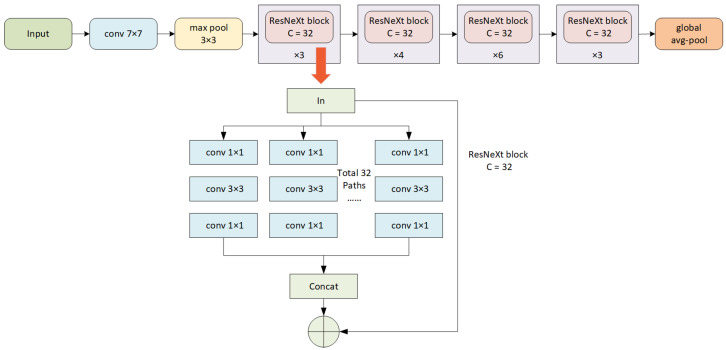
The architecture of the proposed feature extractor based on ResNeXt50.

**Figure 5 sensors-23-09217-f005:**
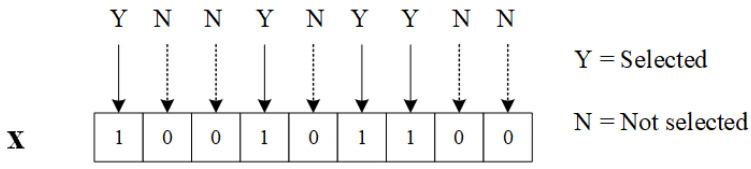
A binary encoding 9-dimensional vector x.

**Figure 6 sensors-23-09217-f006:**
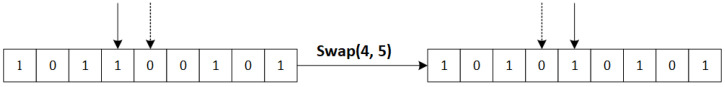
Illustration of swap transformation.

**Figure 7 sensors-23-09217-f007:**
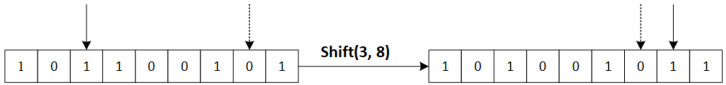
Illustration of shift transformation.

**Figure 8 sensors-23-09217-f008:**
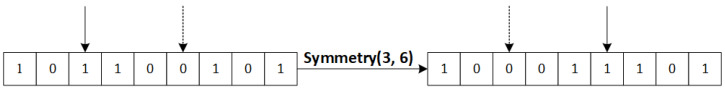
Illustration of symmetry transformation.

**Figure 9 sensors-23-09217-f009:**
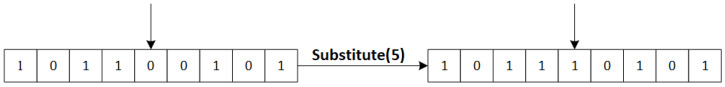
Illustration of substitute transformation.

**Figure 10 sensors-23-09217-f010:**
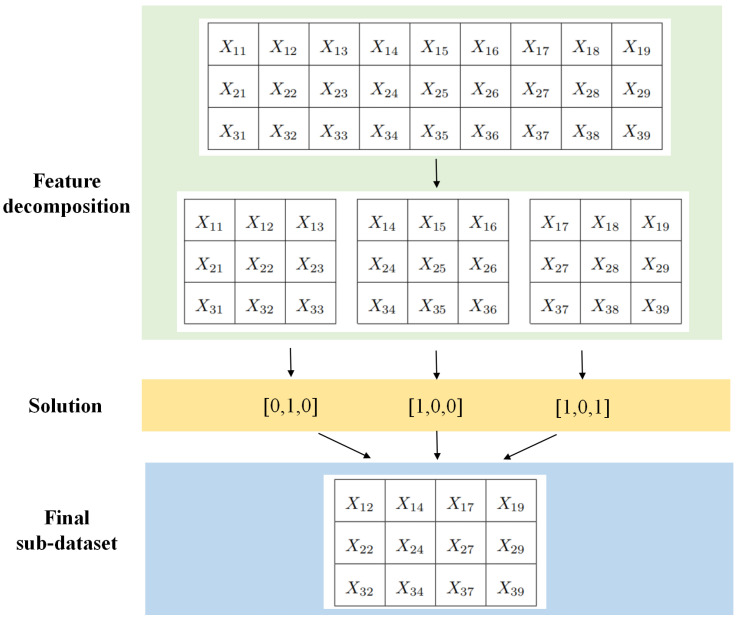
Feature decomposition when data is decomposed into three sub-datasets.

**Figure 11 sensors-23-09217-f011:**
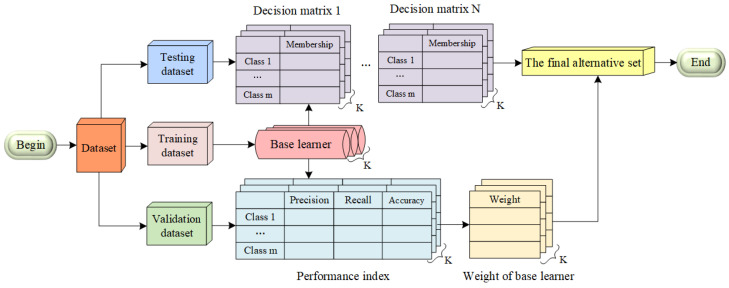
The flowchart for ensemble learning method based on group decision making.

**Figure 12 sensors-23-09217-f012:**
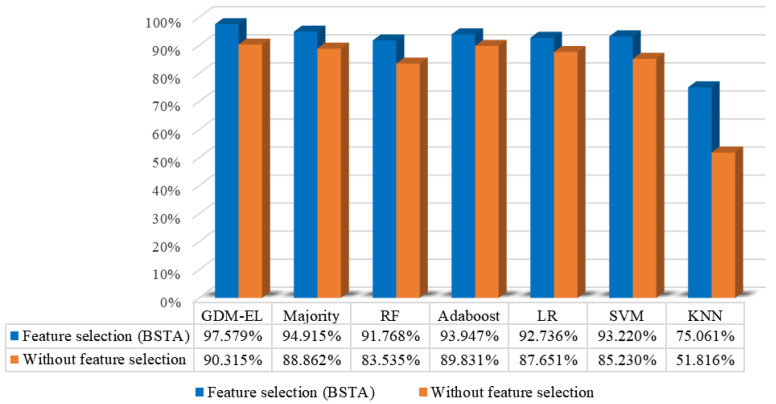
Ablation experiment: Recognition performance of EL methods and base learners. The results above are based on features extracted by ResNeXt50. The recognition accuracy without feature selection and the recognition accuracy with feature selection method based on BSTA are in orange and blue respectively.

**Figure 13 sensors-23-09217-f013:**
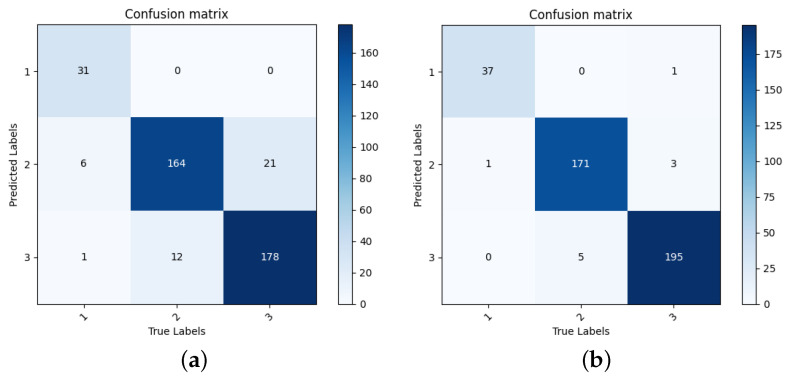
Confusion matrices of GDM-EL for sintering state recognition. (**a**) GDM-EL without feature selection; (**b**) GDM-EL with fearure selection.

**Table 1 sensors-23-09217-t001:** Classification result confusion matrix.

Truth	Prediction	
	**Positive Class**	**Negative Class**
Positive Class	TP	FN
Negative Class	FP	TN

**Table 2 sensors-23-09217-t002:** Comparison of GDM-EL and majority vote.

Method	Calculation Formula
Majority vote	$Q^{*}=Q_{\underset{j}{\arg\max}\sum_{k=1}^{K}\frac{1}{K}\cdot d_{j}^{k}}$
GDM-EL	$Q^{*}=Q_{\underset{j}{\arg\max}\sum_{k=1}^{K}W_{k}d_{j}^{k}}$

**Table 3 sensors-23-09217-t003:** Parameter settings for feature selection based on BSTA.

Population	SE	pct	p	q	Number of Sub-Datasets	Total Rounds
50	20	0.5	0.9	0.1	4	10

**Table 4 sensors-23-09217-t004:** Description of the sintering dataset obtained by feature extraction and feature selection.

Initial Features Extracted by ResNeXt	Features Selected Using BSTA	Samples	Classes
512	133	1651	3

**Table 5 sensors-23-09217-t005:** Description of public datasets.

Datasets	Samples	Classes	Attributes
Letter image	20,000	26	16
Blocks Classification	5473	5	10
Dry Bean	13,611	7	17
Musk	6598	2	168
Isolet	7797	26	617
Pen Digits	10,992	10	16
Waveform	5000	3	21

**Table 6 sensors-23-09217-t006:** Classification accuracy(%) for each combination strategy on different datasets.

Datasets	$W_{k}^{1}=P_{k}+R_{k}+A_{k}$	$W_{k}^{2}=P_{k}R_{k}A_{k}$	$W_{k}^{3}=\frac{3}{\frac{1}{P_{k}}+\frac{1}{R_{k}}+\frac{1}{A_{k}}}$
Letter image	87.863	**88.338**	87.875
Blocks	94.386	**95.753**	95.345
Dry Bean	90.046	**90.689**	90.284
Musk	**97.120**	96.139	96.970
Isolet	93.716	**93.844**	92.885
Pen Digits	96.884	**97.100**	96.884
Waveform	85.750	85.600	**86.150**

**Table 7 sensors-23-09217-t007:** Weights of base learners.

	Adaboost	RF	KNN	LR	SVM
Non-FS	0.316848	0.23589	0.067454	0.125859	0.253953
FS-BSTA	0.25157	0.23587	0.121407	0.173229	0.217915

’Non-FS’ shows the weights of base learners without feature selection; ’FS-BSTA’ shows the weights with feature selection method based on BSTA.

**Table 8 sensors-23-09217-t008:** Performance comparison of different methods for sintering state recognition.

	VGG16	Inception-v3	ResNet50	ResNeXt50	Fine-Tuning	Proposed Method
Accuracy(%)	85.714	88.377	87.409	90.073	92.252	**97.579**
F1-score	0.820	0.857	0.834	0.885	0.911	**0.975**

## Data Availability

Due to privacy or ethical restrictions, data is unavailable.

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
