# Peer review of "Deep Learning Based Feature Selection and Ensemble Learning for Sintering State Recognition"

_sensors, 2023, doi:10.3390/s23229217_

Round 1
Reviewer 1 Report
Comments and Suggestions for Authors
addresses a critical issue in the iron ore sintering process and presents a novel solution that has the potential to revolutionize the industry. The study focuses on improving the quality control of the sintering process, which is of paramount importance in the blast furnace ironmaking process.
One of the key strengths of this paper is its ability to bridge the gap between traditional, intuition-based quality assessment and cutting-edge technology. The authors rightly point out that relying solely on operators' visual observations of sinter cross sections is prone to inaccuracies due to external factors and operator experience. By introducing deep learning and ensemble learning techniques, the paper proposes a systematic and data-driven approach to sintering state recognition.
The methodology employed in this study is comprehensive. The use of ResNeXt for feature extraction from infrared thermal images is a wise choice, as deep learning models have shown remarkable success in image analysis tasks. Furthermore, the incorporation of the binary state transition algorithm (BSTA) for feature selection ensures that only relevant and meaningful features are retained, eliminating noise and redundancy. The ensemble learning approach, based on group decision making (GDM), adds an additional layer of sophistication to the recognition process.
The paper's emphasis on industrial experimentation and validation is OK. The fact that the proposed method has been tested in a real-world steel plant setting lends credibility to its practical applicability. The authors' assertion of the method's effectiveness and superiority is substantiated by the experimental results, which indicate improved recognition accuracy. The contribution is not bad. The paper is well-written, although some typos are found. It should be revised again.
However, there are a few areas where the paper could be enhanced:
Clarity in Presentation: The paper's technical content is sound, but some sections may benefit from improved clarity in presentation. Providing more intuitive explanations for complex concepts and algorithms could make the paper more accessible to a broader audience.
Visual Aids: Incorporating visual aids such as diagrams, charts, or sample images could help readers better understand the proposed methodology and the sintering process itself.
Literature review: The literature reviews should be seriously added. It should be explained more about vision-based object recognition. For example, the following papers using vision-based object/face/gesture detection must be added and discussed.
- Anime face recognition to create awareness. Int. j. inf. tecnol. (2023). https://doi.org/10.1007/s41870-023-01391-8
- A Review of Computer-Based Gesture Interaction Methods for Supporting Disabled People with Special Needs. ICCHP (2) 2016: 503-506
- 3D Human Motion Analysis for Reconstruction and Recognition. Articulated Motion and Deformable Objects. AMDO 2014. Lecture Notes in Computer Science, vol 8563. Springer, Cham. https://doi.org/10.1007/978-3-319-08849-5_12
Future Work: While the paper focuses on the immediate application of the proposed method, briefly discussing potential future directions or broader implications of the research could add depth to the conclusion.
In conclusion, this paper is a well-conceived and executed study with significant potential for impact in the field of iron ore sintering. By leveraging advanced technologies, the authors have taken a crucial step towards improving the accuracy and reliability of sintering quality assessment. With some suitable improvements in presentation and additional insights into future work, this paper can serve as a valuable resource for researchers and professionals in the metallurgy industry. The paper also could be strengthened by providing a thorough review of related work. With these improvements, the paper will have a broader impact and be more useful to the research community. Therefore, I recommend that this paper could be accepted after proper changes. If not, I am afraid to reject this paper.
addresses a critical issue in the iron ore sintering process and presents a novel solution that has the potential to revolutionize the industry. The study focuses on improving the quality control of the sintering process, which is of paramount importance in the blast furnace ironmaking process.
One of the key strengths of this paper is its ability to bridge the gap between traditional, intuition-based quality assessment and cutting-edge technology. The authors rightly point out that relying solely on operators' visual observations of sinter cross sections is prone to inaccuracies due to external factors and operator experience. By introducing deep learning and ensemble learning techniques, the paper proposes a systematic and data-driven approach to sintering state recognition.
The methodology employed in this study is comprehensive. The use of ResNeXt for feature extraction from infrared thermal images is a wise choice, as deep learning models have shown remarkable success in image analysis tasks. Furthermore, the incorporation of the binary state transition algorithm (BSTA) for feature selection ensures that only relevant and meaningful features are retained, eliminating noise and redundancy. The ensemble learning approach, based on group decision making (GDM), adds an additional layer of sophistication to the recognition process.
The paper's emphasis on industrial experimentation and validation is OK. The fact that the proposed method has been tested in a real-world steel plant setting lends credibility to its practical applicability. The authors' assertion of the method's effectiveness and superiority is substantiated by the experimental results, which indicate improved recognition accuracy. The contribution is not bad. The paper is well-written, although some typos are found. It should be revised again.
However, there are a few areas where the paper could be enhanced:
Clarity in Presentation: The paper's technical content is sound, but some sections may benefit from improved clarity in presentation. Providing more intuitive explanations for complex concepts and algorithms could make the paper more accessible to a broader audience.
Visual Aids: Incorporating visual aids such as diagrams, charts, or sample images could help readers better understand the proposed methodology and the sintering process itself.
Literature review: The literature reviews should be seriously added. It should be explained more about vision-based object recognition. For example, the following papers using vision-based object/face/gesture detection must be added and discussed.
- Anime face recognition to create awareness. Int. j. inf. tecnol. (2023). https://doi.org/10.1007/s41870-023-01391-8
- A Review of Computer-Based Gesture Interaction Methods for Supporting Disabled People with Special Needs. ICCHP (2) 2016: 503-506
- 3D Human Motion Analysis for Reconstruction and Recognition. In: Perales, F.J., Santos-Victor, J. (eds) Articulated Motion and Deformable Objects. AMDO 2014. Lecture Notes in Computer Science, vol 8563. Springer, Cham. https://doi.org/10.1007/978-3-319-08849-5_12
Future Work: While the paper focuses on the immediate application of the proposed method, briefly discussing potential future directions or broader implications of the research could add depth to the conclusion.
In conclusion, this paper is a well-conceived and executed study with significant potential for impact in the field of iron ore sintering. By leveraging advanced technologies, the authors have taken a crucial step towards improving the accuracy and reliability of sintering quality assessment. With some suitable improvements in presentation and additional insights into future work, this paper can serve as a valuable resource for researchers and professionals in the metallurgy industry. The paper also could be strengthened by providing a thorough review of related work. With these improvements, the paper will have a broader impact and be more useful to the research community. Therefore, I recommend that this paper could be accepted after proper changes. If not, I am afraid to reject this paper.
Reviewer 2 Report
Comments and Suggestions for Authors
This paper demonstrates a sintering state recognition method based on feature selection and ensemble learning. Many experiments are employed to verify the effectiveness of the proposed method. I suggest the authors address the following points.
1. Many image recognition methods based on deep learning have already been presented, such as AlexNet, VGG, and Inception. Is this method competitive compared to a large number of popular image recognition methods? The comparison experiments with popular recognition methods should be added.
2. In Algorithm 1, the author stated “the termination condition is met”, the termination condition should be specifically described.
3. In section 3, the authors describe the related work and the detailed methods. It cannot clearly express which one is the proposed method.
4. The paper should clearly describe the details of the proposed method.
5. In the experiment part, only one metric (precision) is employed to evaluate the performance of the method. The author should use more performance measures, such as recall and F1-score.
6. The “conclusion” part could be strengthened. The main conclusion and contribution should be clearly introduced, and the unnecessary expressions should be deleted.
7. The English of the whole paper should be improved.
Reviewer 3 Report
Comments and Suggestions for Authors
Review of the article The article proposes an automated method for assessing the sintering state of iron ore. In this study, a new sintering state recognition method using deep learning based feature selection and ensemble learning is proposed. A pretrained deep neural network ResNeXt50 is used to extract features from the infrared thermal images of sinter cross section at the tail of the sintering machine. The research results obtained may be of interest to specialists in materials science, technology for the production of cast iron, steels and alloys. There are some minor comments regarding the work. The experimental results of the study are not discussed fully enough. How many replicate experiments were performed? It is not clear how the method’s efficiency accuracy value of 97.579% was obtained?
Round 2
Reviewer 1 Report
Comments and Suggestions for Authors
The revised manuscript has been significantly better. Thus, I recommend accepting this manuscript.
Reviewer 2 Report
Comments and Suggestions for Authors
I recommend the paper for publication.